# Dental Anomalies in Rare, Genetic Ciliopathic Disorder—A Case Report and Review of Literature

**DOI:** 10.3390/ijerph17124337

**Published:** 2020-06-17

**Authors:** Tamara Pawlaczyk-Kamieńska, Hanna Winiarska, Tomasz Kulczyk, Szczepan Cofta

**Affiliations:** 1Department of Risk Group Dentistry, Poznan University of Medical Sciences, Bukowska 70, 60-812 Poznań, Poland; 2Department of Pulmonology, Allergology and Respiratory Oncology, Poznan University of Medical Sciences, Szamarzewskiego 82/84, 60-569 Poznań, Poland; winiarskahanna@ump.edu.pl (H.W.); scofta@ump.edu.pl (S.C.); 3Section of Dental Radiology, Department of Biomaterials and Experimental Dentistry, Poznan University of Medical Sciences, Bukowska 70, 60-812 Poznań, Poland; tkulczyk@ump.edu.pl

**Keywords:** primary ciliary dyskinesia, developmental dental anomalies, dental morphology

## Abstract

Background: Primary ciliary dyskinesia (PCD) is a rare, ciliopathic disorder. In many ciliopathies, dental anomalies are observed alongside other symptoms of the disease. To date, there are no published reports concerning the dental developmental problems that are associated with ciliary defects in PCD patients. Methods: Patients suffering from PCD underwent dental clinical examination, which included the assessment of developmental disorders regarding the number and morphological structure of the teeth (size and shape) as well as developmental disorders of mineralised dental tissues. Then, three-dimensional radiographic examination was performed utilising Cone Beam Computed Tomography (CBCT). Results: Four PCD patients, aged 31-54, agreed to enter the study. Dental examinations showed the presence of dental developmental disorders in three of them. Additionally, CBCT showed abnormalities in those patients. Conclusions: 1. The dental phenotype in PCD patients seems to be heterogeneous. Tooth developmental disorders resulting from abnormal odontogenesis may be a symptom of PCD that is concomitant with other developmental abnormalities resulting from malfunctioning primary cilia. 2. Patients with ciliopathies are likely to develop dental developmental defects. Therefore, beginning in early childhood, they should be included in a targeted specialised dental programme to enable early diagnosis and to ensure dedicated preventive and therapeutic measures.

## 1. Introduction

Primary ciliary dyskinesia (PCD) is a rare, autosomal recessive genetic disease. It is less often seen through autosomal dominant or gender-related inheritance [1]. PCD affects ∼1:20000 individuals, with a reported prevalence of 1:4000 to <1:50000 [2]. The signs and symptoms of this condition are caused by abnormal ciliary structure and function in the ciliated epithelium lining of the nose; sinuses; airways; Eustachian tube; middle ear; fallopian tube; and flagellum of sperm cells. Along with other diseases that are caused by defects of cilia, PCD belongs to the group of disorders known as ciliopathies [1]. The most serious symptoms of PCD result from impairment of the upper and lower respiratory tracts’ epithelium [1,3]. This disease is also a cause of infertility in men, given that the defect in structure also affects sperm cells [1]. In about half of PCD patients, apart from sinusitis and bronchiectasis, visceral inversion (situs inversus) also occurs, which shows the impact of abnormal function of cilia on embryogenesis. This triad of symptoms, called Kartagener Syndrome, was first described in 1904 [1,3].

In many ciliopathies, one of the symptoms may be developmental dental anomalies [4,5,6,7,8,9,10,11]. Abnormalities in the number of teeth (an increase or decrease), tooth size (microdontia, macrodontia), anatomical structure, teeth location, and mineralised hard dental tissues (enamel hypoplasia, dental hypoplasia) have been described [4,5,6,7,8,9,10,11,12]. The available literature on the dentition of PCD patients is poor. To date, one case describing unusual dental morphology in a patient with Kartagener Syndrome has been presented [13].

The aim of this study is the clinical and visual evaluation of developmental teeth disorders in patients with PCD.

## 2. Material and Methods

The research was conducted in accordance with the ethical principles of the World Medical Association Declaration of Helsinki [14] and was approved by the Ethical Committee of the Poznan University of Medical Sciences, Poland (NO. 371/20). The study included patients from the Department of Pulmonology, Allergology and Respiratory Oncology, Poznan University of Medical Sciences, who were suffering from PCD.

Clinical dental examinations were conducted by two dentists in a dental surgery setting and included the assessment of developmental disorders regarding the number and morphological structure of the teeth (size and shape) as well as developmental disorders of mineralised dental tissues. Prior to the clinical examination, the examiners were calibrated (k = 0.85). The study included all teeth in the oral cavity, except for the third molars, which commonly show diverse structures. To determine the condition of the dental pulp, a thermal test (cold reaction with ethyl chloride) was used. The next stage was to assess the presence of enamel developmental defects on all tooth surfaces using the modified Developmental Defects of Enamel index [15].

For every patient, an X-ray examination was performed to assess the remaining craniofacial morphological parameters and to exclude the presence of possible pathologies. Three-dimensional radiographic examination was performed utilising Cone Beam Computed Tomography (CBCT). A Scanora 3D XL CBCT unit (Soredex Co., Tuusula, Finland) using a low-emission exposure protocol (89kV, 6mA, 16s, voxel 0.2) was used. Radiographic data were reconstructed and visualised through OnDemand3D software (Cyber Med, Seoul, Korea).

## 3. Results

Thirteen patients under the care of the Department of Pulmonology, Allergology and Respiratory Oncology, Poznan University of Medical Sciences were identified as potential participants. Four patients, upon presentation of the test procedure, agreed in writing to enter the study. The patients ranged in age from 31 to 54 years and included three women and one man. For all four patients, the diagnosis of PCD was based on a typical clinical picture, which was confirmed by examination of their bronchial mucosa bioptate under an electron microscope.

Dental examinations showed the presence of dental developmental disorders in all of the four patients. In three of them, there were present enamel defects (white creme demarcated opacities, diffused opacities or hypoplasia). The defects were present only in one group of teeth (incisors), or all the teeth were affected. Moreover, in two patients the clinical examination revealed abnormalities in tooth morphology: local microdontia or congenitally missing teeth. The defect was observed in upper lower incisors.

The CBCT study showed dental abnormalities in three of patients. In one patient (patient 1), CBCT showed mesial dilaceration of the upper second premolar roots (teeth 15 and 25), lingual dilaceration of the right mandibular canine root (43) (Figure 1), and shortening of the root of the central incisors (teeth 11 and 21). In addition, the vertical dimension of the roots of teeth 11 and 21 was smaller than the height of their crowns. Therefore, the disorder met the criteria of rhizomicria (the length of the root in tooth 11 was 7mm, and the length of the crown was 11mm (Figure 2); for tooth 21, the measurements were 7.5 and 11, respectively) (Figure 3). The CBCT study of another patient (patient 2) showed a decrease in the vertical dimension of the upper teeth, including the molars, premolars and canines, with the proper structure (Figure 4, Figure 5). There were no abnormalities in the mandibular teeth, although their vertical dimension was within the lower limits. In third patient the CBCT confirmed the clinical examination showing missing tooth.

## 4. Discussion

The process of tooth development (odontogenesis) depends on constant, mutual and alternating interaction (through the production and secretion of regulatory agents, the so-called ‘’signal molecules’’) of cells of two adjacent tissues: ectomesenchyme (neuromesenchyme), which is a foetal connective tissue that develops from neural crests, and the ectodermal epithelial tissue that lines the primary oral cavity [16]. Odontogenesis is controlled and regulated by genetic factors. To date, approximately 300 genes that are responsible for process induction and for interactions between neuromesenchymal and epithelial tissue have been identified [16,17,18]. Most genes encode protein signalling molecules that are associated with individual signalling pathways, which act as local mediators of intercellular signalling [16,17].

Signal pathways are responsible for cell cycle control, including the response to signals of interaction between epithelial and neuromesenchymal tissue as well as to environmental signals [16]. Mutual induction of ectomesenchymal and epithelial cells is a basic factor that enables morphogenetic changes (i.e., giving the appropriate shape of the crowns and roots, which are different for each individual dental group—incisors, canines, premolars and molars) as well as enabling histogenetic changes. The latter would include tissue differentiation in a tooth’s germ together with the formation of mineralised structures (the dentin, enamel and cementum) and nonmineralised structures (the pulp and periodontium) [16,18,19,20].

An indispensable element of the transmission pathways in embryogenesis is the immobile primary cilia [12,21], which are present on the surfaces of almost all cells of the human body. These are short, specialised projections that mediate the reception and transmission of signals thanks to numerous membrane receptors; thus, they play an extremely important role in the development and functioning of most tissues and organs [12,22,23]. Primary cilia are also present on odontoblasts and ameloblasts, where they are involved in regular tooth morphogenesis [12,23,24,25]. Laboratory studies of dental buds in mice carried out by Hampl et al. (2017) [12] found primary cilia at different stages of tooth bud development, both in the cells of dental tissues of ectodermal origin and in the mesenchymal tissue cells surrounding the bud.

The process of cilia formation is multistaged and precisely controlled. Mutations of genes that encode the proteins that are responsible for the structure and function of cilia are the cause of their impaired growth and/or maintenance of normal structure, and they may even account for their complete absence. The cilia are present in almost all human cells. If they are defective, they affect many tissues and organs and lead to several diseases in the category of ciliopathies. Specific clinical disorders are related to the role that damaged cilia play physiologically [22]. The group of ciliopathies in which dental abnormalities have been noted include, among others, Bardet–Biedl syndrome [4,5], oral-facial-digital syndrome, [6], Ellis van Creveld syndrome [7,8], Weyers acrofacial dysostosis (Curry Hall syndrome) [9], Joubert syndrome [10] and Cranioectodermal dysplasia (Sensenbrenner syndrome) [11]. The abnormalities of ciliopathies that are described in the literature concern an increase or decrease in the number of teeth, changes in their shape and size, as well as anomalies in mineralised tooth tissues.

To date, one case of atypical morphology and developmental enamel defects has been described in a patient (a 13-year-old female) with Kartagener Syndrome [13]. On clinical examination, she was found to have mixed dentition, and enamel linear hypoplasia was noted on her permanent teeth. Moreover, all her lower incisors and upper central incisors had an aberrant shape with notching. The crown of her right lateral maxillary incisor (12) was diminutive in size, and the left lateral incisor (22) was congenitally missing, which was also confirmed by radiological examination. Finally, tooth 12 had a diminutive root.

Irreversible enamel defects are the most common developmental disorder found in dental practice [15,26,27]. Enamel biology is a part of integrated system biology and is subject to the same physiological anomalies. Factors that may interfere with amelogenesis include genetic factors and systemic diseases that can have adverse effects (either directly or indirectly) through complications of applied pharmacotherapy. It may comprise a single tooth, a group of teeth or the entire dentition, and it may present as a white-cream or yellow-brown opalescence, hypoplasia (depressions, transverse or vertical lines), discoloration, the complete absence of enamel or other multiple and complex defects [15,27]. Enamel defects of varying severity were found in three of the four patients examined. The aetiology of these disorders in PCD patients has not yet been clarified. This is perhaps a side effect of either chronic antibiotic therapy from an early age (i.e., during the amelogenesis period) or complications of the underlying disease. The theory that the disorder is caused by a so-far undetected genetic defect (possibly associated with a primary ciliary defect) resulting in incorrect signal transmission in one or more signal pathways is also likely.

The least frequently observed problem is the malformation of dental roots, although this issue may be underestimated; in many cases, routine clinical examination does not detect them. Diagnostics is only possible via a radiological examination. In many cases, the structure of the crowns is normal, and the changes affect the roots only. The aetiology of these abnormalities is probably multifactorial, and it has yet to be fully explained [20]. Improper signals during root development may cause a disturbance of the developmental pattern, leading to morphological abnormalities of the roots; the presence of supernumerary roots or additional structures within the root; pyramidal-shaped roots; and taurodontism [28,29,30]. A malformation may comprise a single tooth or a group of teeth whose root development took place while a given damaging factor was active. In the current research, the participation of the local etiological factor was eliminated based on each patient’s history.

Laboratory tests that were conducted by Hampl et al. (2017) [12] found the primary cilia to be an important element of cell signalling. Any abnormalities or lack of cilia can interfere with odontogenesis [12,24]. The cases of patients with PCD that were presented in their study and the patient with Kartagener syndrome described by Merrett and Durning (2005) [13] showed that the dental phenotype of this group of patients can vary, ranging from no abnormalities and reducing crown size, shortening the entire tooth length and shortening the roots up to reducing the number of teeth due to missing buds. As Hampl et al. suggested (2017) [12] the individual, unique dental phenotype is probably not associated with a single gene or a selected signal pathway but rather with a specific response of the tooth’s tissues to a complex combination of signals. Such a theory could explain the occurrence in PCD patients of both a variety of developmental disorders and the absence of any examinable dental abnormalities within the same genetically related disease. Only four of thirteen patients, who are under the care of the Department of Pulmonology, Allergology and Respiratory Oncology, agreed to enter the study. The results and the dental phenotype of other PCD patients are unknown. The scarcity of reports on the dentition of this group of patients suggests that further clinical trials are needed.

## 5. Conclusions

1. The dental phenotype in patients with PCD seems to be heterogeneous, and tooth developmental disorders resulting from abnormal odontogenesis may be a symptom that is concomitant with other developmental abnormalities resulting from malfunctioning primary cilia.

2. Patients suffering from ciliopathic-related diseases are likely to develop dental developmental defects. Therefore, beginning in early childhood, they should be included in a targeted specialised dental programme to enable early diagnosis and ensure dedicated and integrated (conservative-orthodontic-prosthetic) preventive and therapeutic measures.

## Figures and Tables

**Figure 1 ijerph-17-04337-f001:**
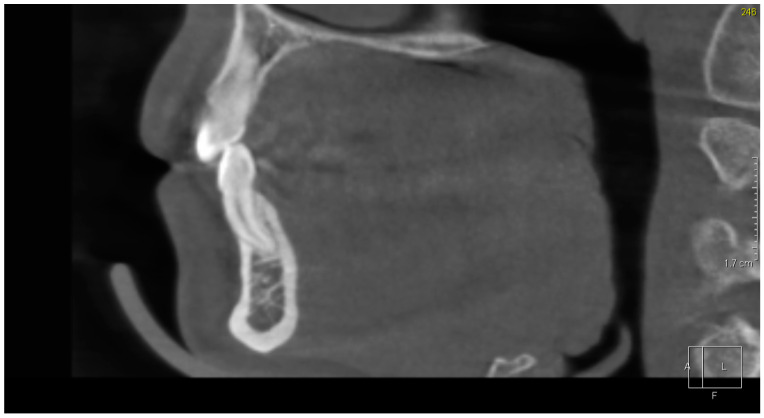
Patient 1. Lingual dilaceration of right mandibular canine (43)—Sagittal reconstruction from Cone Beam Computed Tomography (CBCT).

**Figure 2 ijerph-17-04337-f002:**
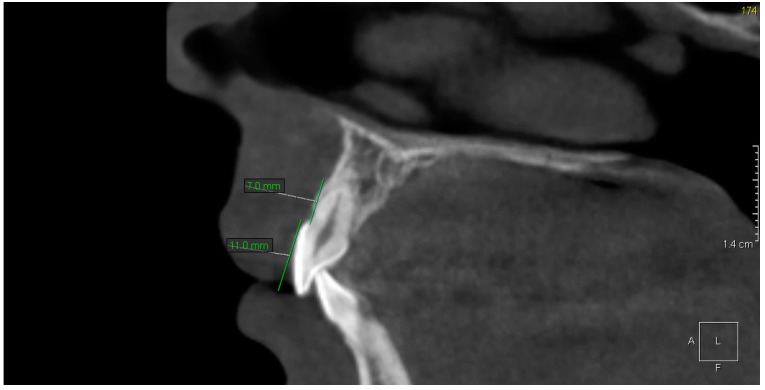
Patient 1. Rhizomicria of upper right central incisor (11)—Sagittal reconstruction from CBCT.

**Figure 3 ijerph-17-04337-f003:**
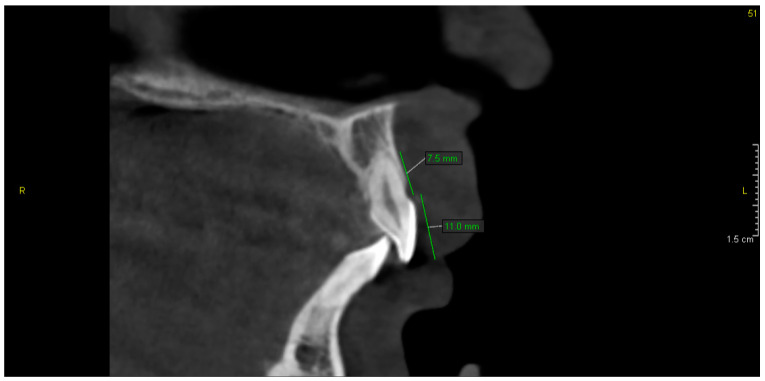
Patient 1. Rhizomicria of upper left central incisor (21)—Sagittal reconstruction from CBCT.

**Figure 4 ijerph-17-04337-f004:**
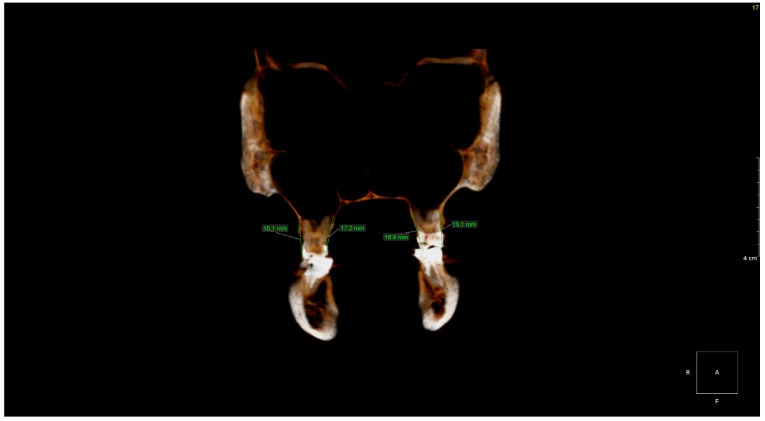
Patient 2. Vertical dimensions of upper first permanent molars (16 and 26)—Coronal volumetric reconstruction from CBCT.

**Figure 5 ijerph-17-04337-f005:**
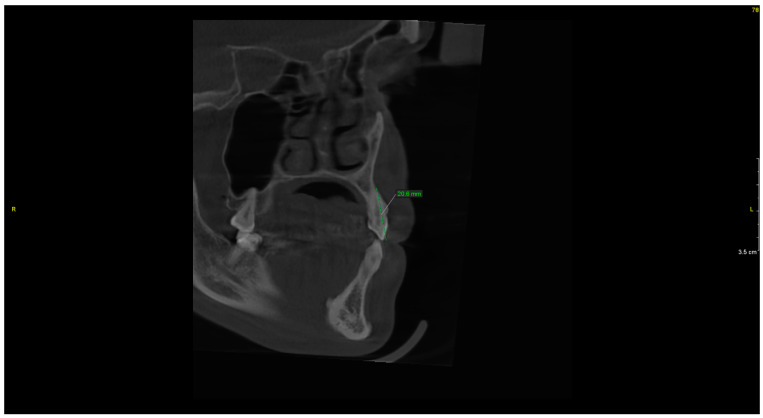
Patient 2. Vertical dimension of upper left canine (23)—Sagittal reconstruction from CBCT.

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
