# Peer review of "Dental Anomalies in Rare, Genetic Ciliopathic Disorder—A Case Report and Review of Literature"

_ijerph, 2020, doi:10.3390/ijerph17124337_

Round 1
Reviewer 1 Report
Dear authors,
I read with pleasure your well written and presented manuscript and minor study.
I only have a few suggestions you may consider for revision.
Introduction
Paragraph one, line 35-41 is quite a long paragraph without any references in between. Therefore I suggest you putting in 1-2 references maybe after the mening line 40 or earlier and absolutely needed one after the sentence (line 40-41) "This disease is also ... etc"(ref)
Results
I would appreciate if you made a comment why patients that said no to the inrollment of the study. Of 13 asked 4 said yes. Please make a comment about the drop outs and possible implications for your study.
Discussion
First paragraph (line 97-105)- at the end of this paragraph (line 105) I am missing a reference or references.
Good Luck with your revision!
Author Response
We would like to thank the reviewer for a careful and thorough reading of this manuscript and invaluable comment. We appreciate the positive feedback from the reviewer. As suggested by the reviewer, we have reviewed carefully the manuscript. The following is our response (the reviewer’s comment is in italics):
Paragraph one, line 35-41 is quite a long paragraph without any references in between. Therefore I suggest you putting in 1-2 references maybe after the mening line 40 or earlier and absolutely needed one after the sentence (line 40-41) "This disease is also ... etc"(ref)
Response:
We fully agree with the reviewer comment. Thank you very much for pointing out this problem.
We have revised the manuscript and added references.
Results
I would appreciate if you made a comment why patients that said no to the inrollment of the study. Of 13 asked 4 said yes. Please make a comment about the drop outs and possible implications for your study
Response:
We have revised our article, and by adding the sentence to the discussion section we have rewritten the manuscript as follows:
Only four of thirteen patients, who are under the care of the Department of Pulmonology, Allergology and Respiratory Oncology, agreed to enter the study. It is not known the possible results, the dental phenotype, of other PCD patients.
Discussion
First paragraph (line 97-105)- at the end of this paragraph (line 105) I am missing a reference or references.
Response:
Thank you very much for this comment. In the revised paper we have added two references at the end of this paragraph.
Reviewer 2 Report
The authors discuss the subject of tooth abnormalities in the course of cliliopathic disorders. Reading the title, I expected an analysis of the factors leading to these disorders and research in a group of children. Maybe observing the erupting teeth and describing emerging disorders. The title should be changed, the authors do not observe development, but state already existing disorders. In addition, I would add a case report and literature review in the title. Only 4 people were evaluated for dental status!
Oral cavity assessment was carried out by two dentists, whether they were calibrated, and what was the compatibility of the results obtained between them. The discussion is very lengthy, sometimes it does not refer to the topic of publication
Author Response
We would like to thank the reviewer for a careful and thorough reading of this manuscript and invaluable comments and constructive suggestions. Please find our point-by-point responses (the reviewer’s comments are in italics).
The authors discuss the subject of tooth abnormalities in the course of cliliopathic disorders. Reading the title, I expected an analysis of the factors leading to these disorders and research in a group of children. Maybe observing the erupting teeth and describing emerging disorders. The title should be changed, the authors do not observe development, but state already existing disorders. In addition, I would add a case report and literature review in the title. Only 4 people were evaluated for dental status!
Response:
Thank you very much for this comment. We agree with the reviewer. In the revised paper we have changed the title:
Dental anomalies in rare, genetic ciliopathic disorder – a case report and review of literature
Oral cavity assessment was carried out by two dentists, whether they were calibrated, and what was the compatibility of the results obtained between them.
Response:
Thank you very much for this vulnerable comment. We have carefully revised the Material and Methods section and we have adad the following sentence:
Prior to the clinical examination, the examiners were calibrated (k=0.83).
The discussion is very lengthy, sometimes it does not refer to the topic of publication
Response:
Thank you very much for this comment. The reviewer is absolutely right – maybe the discussion section is too long. We have revised this part of our manuscript.